# Comparing Classical and Bayesian Panel Kink Regression Frameworks in Estimating the Impact of Economic Freedom on Economic Growth

Emmanuel Mensaklo [1,2,*], Chukiat Chaiboonsri [2], Kanchana Chokethaworn [2] and Songsak Sriboonchitta [2]

1  School of Business, Evangelical Presbyterian University College, Ho P.O. Box HP 678, Ghana
2  Faculty of Economics, Chiang Mai University, Chiang Mai 50200, Thailand; chukiat.chai@cmu.ac.th (C.C.); kanchana.ch@cmu.ac.th (K.C.); songsak.s@cmu.ac.th (S.S.)
*  Correspondence: emmanuel_mensaklo@cmu.ac.th or emmanuel.mensaklo@epuc.edu.gh

**Abstract:** This study aims to accomplish three main tasks. Firstly, it seeks to determine the more appropriate choice between classical and Bayesian methods in estimating a pooled panel kink regression model under the condition of a known but bounded policy variable choice that serves as a kink point. Secondly, as a product of the first target, the study seeks to provide empirical evidence for the economic growth–economic freedom nexus in five top-performing economies in Sub-Saharan Africa. Using index explanatory variables, which are bounded between 0 and 100, and using both numerical and graphical methods, the findings show that the use of the Bayesian method is more appropriate in characterizing the data than the classical OLS framework, as the former better accounts for randomness via the use of posterior distributions. Finally, the study further employed both threshold and Bayesian pooled panel kink regressions, with mixed results. The Bai–Perron test confirmed that the economic freedom index has a single threshold value of 56.70. Whereas the threshold estimates show a negative impact of economic freedom on growth in both low and high regimes, the Bayesian estimates reveal that economic freedom has a negative impact on growth in a low regime but a positive impact in a high regime. Our novel findings show that there exists a nonlinear impact of economic freedom on growth. This provides some guidance and caution in charting policy paths that seek to achieve economic growth.

**Keywords:** Bayesian panel kink regression; economic freedom; economic growth; threshold regression

## 1. Introduction

During the years preceding the emergence of the COVID-19 pandemic, a period which saw almost all economies of the world obtaining substantial negative growth rates, several economies within the Sub-Saharan Africa (SSA) region had recorded moderately to substantially sustained economic growth rates, with several countries emerging as top performers in the region. Unfortunately, this narrative has dramatically changed with only five economies, which are Benin, Côte d'Ivoire, Ethiopia, Rwanda, and Tanzania, exhibiting post-COVID-19 strong recovery from, and resilience to, the residual effects of the pandemic and the ongoing Russia–Ukraine conflict, although most of the negatively affected economies have recorded substantial investments in both physical and human capital during the period under review (AfDB 2023).

It has been reported that, even though the economic meltdown had been widespread across the SSA sub-region, with thirty-one of the economies recording weaker growth rates in 2022 relative to 2021, the sub-region outperformed most world regions in 2022 in relative terms. There are projections that, based on the current state of the region's economic resilience, the aforementioned five of the six pre-pandemic top-performing economies will reclaim their glory in the league of the world's ten fastest-growing economies between 2023

and 2024. For concreteness, whilst most economies within the sub-region hardly could achieve real GDP growth rates beyond 4% in 2021 and 2022, Benin achieved 7.16% and 6%, La Cote d' Ivoire achieved 7.04% and 6.7%, Ethiopia achieved 5.64% and 5.3%, Rwanda achieved 10.88% and 8.20%, and Tanzania achieved 4.28% and 4.7% for the respective years (IMF 2023; AfDB 2023).

Among the many empirical findings, the role of economic freedom in achieving sustainable economic growth within the top-performing economies in the sub-region has gained considerable attention, a role that had been long posited by several economic theories. Several very recent propositions and findings have, however, presented opposing views. Both positions are briefly discussed in the next section.

In its analytical notes on sub-regional economic outlooks, the IMF (2023) has highlighted a growing phenomenon of geo-economic fragmentation of SSA economies over the last two decades. This is a development that the IMF (2023) has noted exposes these economies to severely disadvantaged positions relative to other sub-regions within the global economic space. In the words of the IMF (2023), "Sub-Saharan Africa stands to lose the most in a severely fragmented world compared to other regions, but there could also be potential benefits if fragmentation is limited. It is important for countries to build resilience against likely fallouts from fragmentation and position themselves to benefit from possible changes in trade and capital flow patterns".

Given the above, it is imperative for the various economies to foster strong intra-sub-regional economic cooperation directed at averting the possible fallouts associated with geo-economic fragmentation to ensure continued economic growth and prosperity for their citizenry. Our study derives its motivation from this, and, therefore, we propose to use pooled panel kink regression to estimate the kink effects of economic freedom on economic growth in the selected Sub-Saharan African economies.

We used two main approaches in our empirical analysis. In the first part, we used graphical and numerical approaches to undertake a comparative analysis of classical (specifically, we employed OLS) and Bayesian methods of parameter estimation using a set of index regressors under the condition of known policy targets. This is in line with the common practice in the literature on conventional kink regression constructs. Hansen (2017) noted that this conventional option is very appropriate in many applications that focus on policy-related choices, where the threshold is known and prescribed by a given policy machinery. In the second part, we computationally determined the threshold values and compared parameter estimates from Bayesian and threshold regressions.

Our study seeks to make the following contributions. Although there are countless studies that have used economic freedom to explain economic growth and other productivity measures, there is no evidence of any existing literature on theoretical and empirical applications of kink regression approaches to study the casualimpacts of economic freedom on growth, particularly where the explanatory variables are bounded indexes with policy choices. We suspect that the long-established economic growth–economic freedom nexus that posited a positive impact of the latter on the former has escaped econometric scrutiny. Since all macroeconomic policies and individual choices are always subject to constraints and opportunity costs, it will be of significant economic importance to commence research in this regard. We aim to fill this gap. Also, by undertaking a comparative analysis using classical and Bayesian frameworks, we seek to add to the existing econometric literature as regards which framework will be more suitable for studies of this kind.

The rest of the paper is organized as follows. Section 2 presents the literature review of some theoretical postulates, paradigms, and empirical studies. Section 3 covers data and the methodology used in the study, while Section 4 covers results and discussions. The last section, Section 5, presents the conclusions.

## 2. A Brief Theoretical and Empirical Literature Review

Several theoretical propositions have been propounded to explain the channels or conduits through which various proxies of economic freedom can affect economic outcomes,

including productivity and real economic growth rates. One of the first known formalized theories that implicitly integrated economic freedom into its constructs is the *Schumpeterian growth theory*, named after Schumpeter (1939) who propounded it. This theory of innovation and entrepreneurship, which was capitalistic in shade, postulated that the introduction of new technologies, products, and production methods serve as prime drivers of economic growth. Economic freedom, the theory noted, guarantees economic agents' sovereignty by fostering a competitive environment, incentives, and market openness, promoting entrepreneurial activity and innovation, thereby promoting economic growth.

The *neoclassical growth theory*, which had been popularized largely by the seminal works of Solow (1956, 1957) and Swan (1956), is another strand of the set of theoretical underpinnings that accentuated the importance of capital accumulation, technological progress, and efficient resource allocation in driving economic growth. Economic freedom, with its focus on property rights protection, market competition, and minimal government intervention, explicitly aligns with the principles of the neoclassical growth theory.

One other important theory, which was pioneered by Buchanan and Tullock (1962), is *public choice theory*. This theory employed economic modeling to study how optimal political decision making can be obtained. The theory posits that individuals and policy-makers act in their self-interest and deliberately respond to incentives in ways that maximize these respective interests. It is believed that economic freedom, by limiting the magnitude of government involvement and rent-seeking behavior, can lead to better policy choices and the creation of a conducive environment for economic growth.

*Trade theory* is another relevant theory that offers a compelling rationale for economic freedom. For instance, Krueger (1998) emphasized the existence of a positive relationship between trade liberalization and economic growth and presented the theoretical foundations of trade theories like comparative advantage. His work concisely highlighted the role of economic freedom in promoting international trade.

In addition to the above major theoretical paradigms, there are a few other recent propositions that emphasize how economic freedom could positively affect output measures. Hall and Lawson (2014) and Doucouliagos and Ulubasoglu (2006) presented some related surveys on this. Also, Feenstra (2015) and Mitra et al. (2014) hypothesized how economic freedom could promote growth via international trade. The crust of their propositions explained how trade openness could facilitate foreign direct investments (FDI) in less developed economies. These theorists argued that FDIs usually come with embodiments of transmissible innovations and technologies, which could promote production technology modernization and efficiency. According to The Heritage Foundation (2017) posited that economies that promote economic freedom systemically help mitigate profligate public expenditures of all categories. They argued that the success of this indirectly promotes low tax rates and burdens on economic agents, a situation that can help stimulate private investments and productivity. Some authors have also hypothesized how monetary policy implementation in an environment where economic freedom is sustained can trigger economic growth through moderating factors like relatively low inflation rates and capital accumulation (Gertz and Evers 2020; Baharumshah et al. 2016).

Whilst all the above theoretical perspectives posited a positive impact of economic freedom on output measures, some others postulated opposing views. For instance, some argued that excessive economic freedom that liberalizes international trade and permits unbounded trade openness might be counterproductive in developing economies from many practical perspectives. Some other theorists emphasized that economic freedom that enlarges the private sector uncontrollably and reduces government size can hamper public investment in some critical areas (for example, human and capital investments) of the macroeconomy, especially in developing economies (Zahonogo 2017; Bergh and Nilsson 2010; Carter 2007; Kneller et al. 1999). We now turn to some empirical studies.

Some previous studies have found strong evidence that economic freedom affects economic outcomes positively. For example, Krueger (1998) provided empirical evidence supporting the idea that economic freedom and free trade contribute to efficiency gains,

productivity improvements, and overall economic growth. In a study by Kacprzyk (2016), security of property rights, quality of monetary policy, freedom to trade, and regulatory policies were found to be positively related to economic growth.

However, some studies found contrasting evidence. For example, Santiago et al. (2020) found that economic freedom has a negative impact on the long-run economic growth of Latin American and Caribbean countries. This finding aligns with that of Justesen (2008) and Carlsson and Lundström (2002), who also found that some categories of economic freedom indicators have a negative relationship with growth. Panahi et al. (2014) found that not all economic freedom indicators have a statistically and economically significant impact on growth.

In summary, the above theories and studies revealed sharply contrasting posits and evidence for the impact of economic freedom on economic growth. Whilst some indicated a positive of the former on the latter, others rooted for a negative impact. This observation provides some indication of a possible nonlinear impact of the former on the latter. Our study seeks to fill this gap by empirically providing evidence for this nonlinearity if it does exist.

## 3. Data and Methodology

### 3.1. Data

The study used balanced panel data from five countries (which are Benin, Côte d'Ivoire, Ethiopia, Rwanda, and Tanzania), covering the periods from 2005 to 2022. The selection of these five economies was motivated by their ranking by AfDB (2023) as the top 5 best-performing economies in Sub-Saharan Africa after the peak of the COVID-19 pandemic. It has been reported that, even though the economic meltdown had been widespread across the SSA sub-region, with thirty-one of the economies recording weaker growth rates in 2022 relative to 2021, the sub-region outperformed most world regions in 2022 in relative terms. There were projections that, based on the current state of the region's economic resilience, the aforementioned five of the six pre-pandemic top-performing economies would reclaim their glory in the league of the world's ten fastest-growing economies between 2023 and 2024. For concreteness, whilst most economies within the sub-region hardly could achieve real GDP growth rates beyond 4% in 2021 and 2022, Benin achieved 7.16% and 6%, La Cote d' Ivoire achieved 7.04% and 6.7%, Ethiopia achieved 5.64% and 5.3%, Rwanda achieved 10.88% and 8.20%, and Tanzania achieved 4.28% and 4.7% for the respective years (IMF 2023; AfDB 2023).

What constitutes economic freedom has been variously defined by different authors (see Miller and Kim 2016; The Heritage Foundation 2023). We adopted the definition by The Heritage Foundation (2023), which is the main source of our dataset for the various proxies for economic freedom for the first part of our study. The index is disaggregated into 12 main domains, with a further regrouping under four broad categories. We selected 10 of the indicators across the broad categories based on data availability for the selected periods (see The Heritage Foundation (2023) for detailed definitions of the variables and the methods used to compute each). These categories include the rule of law (property rights and government integrity), government size (tax burden and government spending), regulatory efficiency (business freedom, labor freedom, and monetary freedom), and market openness (trade freedom, investment freedom, and financial freedom). Data on real per capita GDP growth rates were sourced from the World Development Indicators database of the World Bank (2023) and AfDB (2023). At this point, it is imperative to briefly present a key feature of our data for the first part of our analysis. We considered only index explanatory variables, which are bounded between 0 to 100 percent, to explore the possibility of their nonlinear impact on economic growth. These index variables are composite in nature, as they are derived from several sub-indicator variables. We believe that, for lack of any better alternatives, each selected index is a fair quantification of economic freedom and a good measure of private sector sovereignty. These variables also reflect all of the key major determinants of economic growth.

For the second part of the estimation, guided by the neoclassical growth theory, we included the following explanatory variables: capital, measured by the annual growth rate of gross fixed capital formation (fixed capital); foreign direct investment (FDI), proxied by net inflows of FDI as a percent of GDP; labor, measured by labor force participation rate; economic freedom, measured by the overall index of economic freedom (economic freedom); and population, proxied by the real annual population (population) growth rate. The first two were sourced from the World Bank (2023), whilst, respectively, labor, population, and economic freedom were obtained from the International Labour Organization (ILO), the United Nations Conference on Trade and Development, and The Heritage Foundation.

### 3.2. Methodology

#### 3.2.1. The Model

For the purposes of our analysis, our modeling procedure took two distinct approaches. The first part focuses on kink regression with a comparative analysis of Bayesian and classical frameworks using univariate indexes of economic freedom. The second part extended our modeling by incorporating some other explanatory variables suggested in the growth literature, and we undertook a comparative analysis of the impact of economic freedom on economic growth within the broader frameworks of Bayesian kink and the traditional threshold regressions.

#### 3.2.2. Kink Regression

There are several formulations of kink regression models. In this study, we followed Zhang et al. (2017). Our general model is given by

$$Y_{it} = \beta_1^- (X_{1,it} - \delta) + \beta_1^+ (X_{1,it} - \delta) + \beta_j' X_{j,it} + \beta_i + u_{it}, \tag{1}$$

where $i = 1, \ldots, N$ are the cross-sectional units, which are the selected Sub-Saharan African countries in our study. $t = 1, \ldots, T$ is the time index, $X_{j,it}$ is a $d \times 1$ vector of regressors, and $\beta_i$ represents the unobserved heterogeneity of the $i$th individual, which can be correlated with $X_{it} = \left( X_{1,it}, X_{j,it}' \right)'$ with $j = 1, \ldots, J$. $(\beta_1^-, \beta_1^+, \beta_j, \delta)$ are parameters to be estimated, whilst $\delta$ could be exogenously given in some cases (see Hansen (2017)), a situation that is the subject matter of the first part of this study. Let $k = X_{1,it} - \delta$, $(k)_- = \min(k, 0)$, and $(k)_+ = \max(k, 0)$. For values of $X_{1,it}$ for which $X_{1,it} \leq 0$, the slope is given by $\beta_1^-$; otherwise, the slope is given by $\beta_1^+$. It is important to highlight the fact that all regressors are continuous variables. However, as an exception from all other variables, the parameters at the kink point are discontinuous. Specifically, the entire regression function $\left( \beta_1^- (X_{1,it} - \delta) + \beta_1^+ (X_{1,it} - \delta) + \beta_j' X_{j,it} \right)$ is continuous as a function of $X_{1,it}$ and $X_{j,it}$. However, there exists discontinuity at the threshold, or the kink point, $\delta$, with respect to the slope function with respect to $x_1$. Using the least squares loss function, the $\beta s$ and $\delta$ are functionally quadratic in $\beta$. However, it is not differentiable with respect to $\delta$. This point was explored in Equation (5) (Hansen 2017; Li et al. 2022). Let $\beta = \left( \beta_1^-, \beta_1^+, \beta_j' \right)'$ and $X_{it}(\delta) = \left( (X_{1,it} - \delta)_-, (X_{1,it} - \delta)_+, X_{j,it}' \right)'$. Equation (1) can be reparameterized as follows:

$$Y_{it} = X_{it}'(\delta)\beta + \beta_i + u_{it}. \tag{2}$$

Denoting $\iota_T$ as the $T \times 1$ vector, let $Y_i = (Y_{i1}, \ldots, Y_{iT})'$, $X_i(\delta) = \left( X_{i1}'(\delta), \ldots, X_{iT}'(\delta) \right)'$, and $u_i = (u_{i1}, \ldots, u_{iT})'$. In a matrix formulation, model (2) can be rewritten as

$$Y_i = X_i(\delta)\beta + \beta_i \iota_T + u_i \tag{3}$$

Under the assumption of a strict exogeneity of all *Xs*, the parameters $\beta, \delta$ can be obtained by minimizing

$$(\hat{\beta}, \hat{\delta}) = \operatorname*{argmin}_{\beta \in \mathcal{B}, \delta \in \Delta} Q_{NT}(\beta, \delta), \tag{4}$$

$$Q_{NT}(\beta, \delta) = N^{-1} \sum_{i=1}^{N} \|Y_i - X_i(\delta)\beta\|^2_{M_{l_T}} \tag{5}$$

is the least squares loss function with $M_{l_T} = I_T - \iota_T \iota'_T / T$. As indicated above, $Q_{NT}(\beta, \delta)$ is functionally quadratic in $\beta$; however, it is not differentiable with respect to $\delta$. This is the key point for the kink regression. The threshold point, $\delta$, can be estimated using

$$\hat{\delta} = \operatorname*{argmin}_{\delta \in \Delta} Q^c_{NT}(\delta), \tag{6}$$

where $Q^c_{NT}(\delta)$ is the concentrated objective function, defined as

$$Q^c_{NT}(\delta) = Q_{NT}(\hat{\beta}(\delta), \delta) = N^{-1} \sum_{i=1}^{N} \|Y_i - X_i(\delta)\hat{\beta}(\delta)\|^2_{M_{t_T}}. \tag{7}$$

$\beta$ can be estimated using

$$\hat{\beta}(\delta) = \operatorname*{argmin}_{\beta \in \mathcal{B}} Q_{NT}(\beta, \delta) = \left( \sum_{i=1}^{N} X'_i(\delta) M_{l_T} X_i(\delta) \right)^{-1} \left[ \sum_{i=1}^{N} X'_i(\delta) M_{l_T} Y_i \right]. \tag{8}$$

Interested readers may refer to Appendix A for an outline provided by Zhang et al. (2017) on the asymptotic properties of $\beta$ and $\delta$. Using the set of equations in (19) (under Appendix A), it is easy to construct the asymptotic confidence intervals for $\hat{\delta}$ and $\hat{\beta}$ and then construct a *t*-test for $\mathbb{H}^*_0 : \delta = \delta^*$ against $\mathbb{H}^*_1 : \delta \neq \delta^*$, where $\delta^*$ is some presumed threshold point.

In the first part, our study followed the model specification of Zhang et al. (2017), with some major modifications that are required to incorporate the first-stage data-splitting proposal made by Chong (2001) and the policy choice target specified by Hansen (2017) to reflect choices of policy index variables. Based on these considerations, our study employed the following pooled panel kink regression model:

$$Y_{it} = \beta_0 + \beta_1^- (X_{it} - \delta)_- + \beta_1^+ (X_{it} - \delta)_+ + u_{it} \tag{9}$$

where $i = 1, \ldots, N$, $t = 1, \ldots, T$ are, respectively, individual country and time indexes. In our study, $N = 5$ and $T = 18$. Under our assumption of a known kink point $\delta$ for policy considerations (Hansen 2017), we set $\delta$ equal to the empirical mean of the observed pooled values of the specific regressors. Intuitively, this choice reflected the overall best performance of the five selected economies over the period considered for the study. Adjusting the model by the empirical mean, we rewrote Equation (9) as

$$Y_{it} = \beta_0 + \beta_1^- (x_{it})_- + \beta_1^+ (x_{it})_+ + u_{it} \tag{10}$$

It is important to note that the suppression of the unobservable heterogeneities of the selected economies offers a huge policy advantage in terms of policy developments and initiatives that seek to integrate these and other economies into one economic block against geo-economic fragmentations, a situation decried by the IMF (2023) as being a big bottleneck for the Sub-Saharan region in its struggle for sustainable regional cooperation and development. The econometric method chosen, together with the problems that can arise from ignoring the possible economic freedom endogeneity, is also highlighted as factors capable of producing biased results (Kacprzyk 2016). To circumvent this challenge, most researchers use instrumental variables and simultaneous equation regression approaches in the literature. These frameworks are inappropriate for the main focus of our study. However, Maneejuk and Yamaka (2020) indicated that the use of kink regression resolves the endogeneity problem.

Based on our objective, we propose estimation procedures that involve fitting a segmented linear regression model generated from OLS and Bayesian fits using Markov chain Monte Carlo (MCMC).

### 3.2.3. Segmented Pooled OLS Estimation

Our relevant model is specified in Equation (9). In the absence of a kink point, a straightforward minimization of the loss function generated from Equation (9) would produce the OLS estimates. However, with our assumption of the existence of a kink point governed by policy choices, Equation (9) needs to be piecewise minimized jointly. From Equation (9), we denote $Y_{it} = (Y_{i1}, \ldots, Y_{iT})'$, $x_{it} = ((x_{it})_-, (x_{it})_+)'$, and $\beta = (\beta_1^-, \beta_1^+)'$. In a matrix form,

$$Y_{it} = x_{it}'\beta + u_{it}, \quad u_{it} \sim \mathcal{N}\left(0, \sigma^2\right). \tag{11}$$

With a strict exogeneity of the index regressors, our pooled OLS estimator can be obtained by minimizing the loss function

$$\hat{\beta_{ols}} = \underset{\beta \in \mathcal{B}}{\mathrm{argmin}} \frac{1}{NT} \sum_{i=1}^{NT} \|Y_i - x_{it}'\beta\|^2. \tag{12}$$

Our empirical model is expressed as

$$gdp_{it} = \beta_1^- (index_{it} - \delta)_- + \beta_1^+ (index_{it} - \delta)_+ + u_{it}, \tag{13}$$

where $gdp_{it}$ denotes the real GDP growth rate in country $i$ in year $t$, and $index_{it}$ is a specific index for economic freedom for country $i$ in year $t$.

### 3.2.4. Bayesian Estimation Using MCMC

In this subsection, the main model considered was that of Equation (13). We implemented an MCMC sampling procedure to estimate all unknown model parameters with the instrumentality of a Bayesian estimator, which concurrently yielded the associated parameter inferences. Following the normal practice in the Bayesian literature, we formulated the prior distributions for our unknown parameters and then combined the joint density to construct the posterior distribution. Following Park et al. (2011), we used a non-informative prior for the coefficient parameters. The prior for $b$ was specified using the $b_0$ and $V_0$ arguments, which are the prior mean and prior precision, respectively, and we set the prior as an independent multivariate normal distribution, expressed as

$$b \sim \mathcal{N}_K\left(m_0, V_0^{-1}\right)$$

The prior for the nuisance parameter $\sigma^2$ is predicated on two main parameters, $c_0$ and $d_0$. We assumed $u_i$ has zero mean and variance $\sigma^2$, and the nuisance parameter $\sigma^2$ is inverse Gamma distributed with $\sigma^2 \sim IG\left(\frac{c_0}{2}, \frac{d_0}{2}\right)$, which required the use of a conjugate prior. The conditional posterior distribution for each parameter was derived from the product of their respective priors and their Gaussian likelihood functions. The conditional posterior distribution for $b$ is $N\left(m_*, V_*^{-1}\right)$, where

$$m_* = \left[m_0 V_0 + \frac{\sum_{t=1}^{NT} x_t y_t}{\sigma^2}\right] \left[V_0 + \frac{\sum_{t=1}^{NT} x_t x_t^\top}{\sigma^2}\right]^{-1} \text{ and } V_* = \left[V_0 + \frac{\sum_{t=1}^{NT} x_t x_t^\top}{\sigma^2}\right], \text{ and the}$$

conditional posterior distribution for $\sigma^2$ is $IG\left(\frac{c_0 + NT}{2}, \frac{d_0 + NTs^2}{2}\right)$, where

$$s^2 = NT^{-1} \sum_{t=1}^{NT} \left[y_t - b^\top x_t\right]^2.$$

Drawing MCMC iterates via the Gibbs sampling algorithm has the advantage of handling small and larger parameter sizes (Koop 2003). Our sampling was executed using the Gibbs sampler. All computations were executed using the R package of Park et al. (2011). Following Chong (2001), this was feasible since the data were split. Marginal likelihood functions were computed using the method of Chib. We undertook diagnostic assessments using trace and density plots to determine whether the Markov chains have attained convergence to their stationary distributions.

### 3.3. Threshold Regression

We now extend our discussion to consider a comparative analysis of the impact of economic freedom on economic growth within the broader frameworks of Bayesian kink and the traditional threshold regressions. To undertake this, we relaxed the assumption of a known policy choice level of the various economic freedom indexes and extended our original model to include some more regressors, as has been suggested by the neoclassical growth theory. We followed the threshold model in Eviews 12, preserving the notations as we implemented our estimation using it. Let $m$ denote the possible number of thresholds, yielding a total of $m + 1$ regimes or regions; $t$ and $i$ denote time and cross-sectional units, respectively; and $j = 0, 1, \ldots, m$ represent the number of observations that fall within a given region. Our linear regression could be formulated as $y_{it} = X'_{it}\beta + Z'_{it}\delta_j + \epsilon_{it}$, where $\epsilon_{it}$ has the usual assumptions and $Z$ is a vector of regime-dependent independent variables, which afforded us the appealing opportunity to examine the dynamics of these regressors in different regimes, if the regimes exist. $X$ is a vector of regime non-varying independent variables. Let $q_t$ be a given threshold variable of interest subject to a preserved order of strictly increasing threshold values given by $(\gamma_1 < \gamma_2 < \cdots < \gamma_m)$. If $\gamma_j \leq q_t < \gamma_{j+1}$, then we would be in regime $j$. The sufficiency condition to be in a regime $j$ requires that the value of the threshold variable must be at least as large as the $j$-th threshold value but not as large as the $(j + 1)$-th threshold. Alternatively, we are in regime $j$ if we have $\gamma_j \leq q_t < \gamma_{j+1}$. For a two-regime model, we have:

$$y_t = X'_t\beta + Z'_t\delta_1 + \epsilon_t \text{ if } -\infty < q_t < \gamma_1$$
$$y_t = X'_t\beta + Z'_t\delta_2 + \epsilon_t \text{ if } \gamma_1 \leq q_t < \infty$$

where $\gamma_0 = -\infty$ and $\gamma_{m+1} = \infty$. We used an indicator function to reparametrize our model, where

$$\begin{cases} 1_j(q_t, \gamma) = 1, \text{ if true} \\ 1_j(q_t, \gamma) = 0, \text{ otherwise} \end{cases}$$

Using a pooled panel model, and letting $1_j(q_t, \gamma) = 1(\gamma_j \leq q_t < \gamma_{j+1})$, our model could be specified as $y_t = X'_t\beta + \sum_{j=0}^{m} 1_j(q_t, \gamma) \cdot Z'_t\delta_j + \epsilon_t$.

Let the cost function of the above equation be given by $S(\delta, \beta, \gamma) = \sum_{t=1}^{T} \left( y_t - X'_t\beta - \sum_{j=0}^{m} 1_j(q_t, \gamma) \cdot Z'_t\delta_j \right)^2$. The parameters $\delta, \beta, \gamma$ of the above equation can be estimated using nonlinear least squares. Model selection to identify the threshold variable $q_t$ and its value was carried out using Bai–Perron tests of sequentially determined thresholds, which were implemented using Eviews 12. To estimate the Bayesian model, the threshold value estimated was used to split the data. The procedure followed as had been previously outlined. Our empirical model for this part is given by the following:

$$economic\ growth = f(economic\ freedom,\ fixed\ capital, population, labour,\ FDI)$$

## 4. Results and Discussions

### 4.1. Data Analysis

4.1.1. Descriptive Statistics and Panel Unit Root Test for Part 1 Estimations

For the first part of our study, we used 10 index explanatory variables, which are stationary at a level between the 0.01 and 0.1 significance levels. Our dependent variable was the real GDP growth rate. Table 1 presents the descriptive statistics of the variables used in the first part of the study. Following Chong (2001), our index variables were reparametrized using the empirical mean splitting approach as follows: economic growth (EcoG), property right (ProR; X1, X2), government integrity (GovI: X3, X4), tax burden (TaxB: X5, X6), government spending (GovS: X7, X8), business freedom (BusF: X9, X10), labor freedom (LabF: X11, X12), monetary freedom (MonF: X13, X14), trade freedom (TraF: X15, X16), investment freedom (InvF: X17, X18), and financial freedom (FinF: X19, X20). Tables 2 and 3 below, respectively, present descriptive statistics of all variables used in the second part of the study and their panel unit root tests.

**Table 1.** Descriptive Statistics of the Variable.

| Statistic | EcoG | BusF | FinF | GovI | GovS | InvF | LabF | MonF | ProR | TaxB | TraF |
|---|---|---|---|---|---|---|---|---|---|---|---|
| Mean | 6.47 | 53.41 | 43.89 | 31.47 | 84.38 | 50.17 | 59.31 | 73.77 | 34.73 | 74.87 | 66.54 |
| Median | 6.75 | 51.56 | 50 | 29.8 | 85.3 | 50 | 57.75 | 74.65 | 30 | 77.4 | 67.2 |
| Maximum | 12.55 | 76.9 | 70 | 67.9 | 93 | 80 | 91.4 | 86.4 | 76.5 | 80.8 | 80.8 |
| Minimum | −5.37 | 40.2 | 20 | 10 | 61.3 | 20 | 38.8 | 54.3 | 20 | 52 | 49.4 |
| Std. Dev. | 3.11 | 8.18 | 13.55 | 10.15 | 5.86 | 16.21 | 11.18 | 6.73 | 10.44 | 7.1 | 6.7 |
| Skewness | −0.84 | 0.66 | −0.49 | 1.29 | −1.05 | −0.15 | 1.24 | −0.53 | 2.46 | −1.98 | −0.13 |
| Kurtosis | 4.89 | 2.84 | 2.71 | 5.38 | 4.91 | 2.07 | 4.11 | 2.94 | 8.91 | 6.52 | 2.38 |

**Table 2.** Descriptive Statistics of the Variables.

| Statistic | Economic Growth | Economic Freedom | Fixed Capital | Population Growth | Labor Force | Net FDI |
|---|---|---|---|---|---|---|
| Mean | 6.4988 | 57.2845 | 10.5845 | 2.6902 | 71.6349 | 2.1349 |
| Median | 6.75 | 56.75 | 9.3883 | 2.7037 | 2.7037 | 1.8768 |
| Maximum | 12.6 | 71.1 | 46.5112 | 3.4768 | 3.4768 | 5.6637 |
| Minimum | −4.9 | 49.4 | −19.8463 | 2.0554 | 2.0554 | −0.1931 |
| Std. Dev. | 3.1754 | 4.8055 | 12.4319 | 0.3013 | 0.3013 | 1.3219 |
| Skewness | −0.7523 | 0.7843 | 0.5269 | −0.0752 | −0.0752 | 0.6888 |
| Kurtosis | 4.2967 | 3.4231 | 3.9781 | 2.9067 | 2.9067 | 2.9572 |
| Jarque–Bera | 14.7957 | 9.899 | 7.753 | 0.1175 | 0.1175 | 7.1245 |
| Probability | 0.0006 | 0.007 | 0.0207 | 0.9429 | 0.9429 | 0.0283 |
| Sum | 584.9 | 5155.61 | 952.6119 | 242.1189 | 242.1189 | 192.1412 |
| Sum Sq. Dev. | 897.4099 | 2055.329 | 13,755.17 | 8.0798 | 8.0798 | 155.5224 |

Panels A, B, and C in Table 3 below display the results of three separate panel unit tests. The results from the Levin, Lin, and Chu test and PP–Fisher chi-square test indicated that all variables are strongly stationary between 0.01 and 0.10 levels, with the exclusion of both intercept and linear trend terms. Also, the Im, Pesaran, and Shin test showed that, with only the intercept term, all variables are strongly stationary between 0.01 and 0.10 significance levels, except population. However, with both the intercept and linear trend terms included, all variables are strongly stationary at 0.05 level, except economic freedom. Therefore, we failed to accept the null hypothesis of the existence of both a common unit root and individual unit root.

**Table 3.** Panel Unit Root Tests on the Variables.

Panel A. Levin, Lin, and Chu Test: Null: Unit root (assumes common unit root process)

| | **With Intercept** | | **With Intercept and Trend** | | **No Intercept and Trend** | |
|---|---|---|---|---|---|---|
| Variable Name | Statistic | Prob | Statistic | Prob | Statistic | Prob |
| Economic growth | −2.7496 | 0.003 | −4.92852 | 0.0000 | −5.4563 | 0.0000 |
| Economic Freedom | −7.53633 | 0.0000 | 1.59457 | 0.9446 | −10.7631 | 0.0000 |
| Fixed Capital | −4.35385 | 0.0000 | −3.88142 | 0.0001 | −3.64411 | 0.0001 |
| Population | −1.3716 | 0.0851 | −1.1025 | 0.0976 | −2.18493 | 0.0144 |
| Labor force | −0.99086 | 0.1609 | 27.6564 | 1.0000 | −5.65988 | 0.0000 |
| FDI | −1.50306 | 0.0664 | 0.65114 | 0.7425 | −7.60399 | 0.0000 |

Panel B. PP–Fisher Chi-square Test: Null: Unit root (assumes individual unit root process)

| | **With Intercept** | | **With Intercept and Trend** | | **No Intercept and Trend** | |
|---|---|---|---|---|---|---|
| Variable Name | Statistic | Prob | Statistic | Prob | Statistic | Prob |
| Economic growth | 41.1822 | 0.0000 | 53.7973 | 0.0000 | 17.6994 | 0.0603 |
| Economic Freedom | 40.5331 | 0.0000 | 31.2738 | 0.0005 | 64.5698 | 0.0000 |
| Fixed Capital | 70.2809 | 0.0000 | 82.9391 | 0.0000 | 42.7531 | 0.0000 |
| Population | 4.83518 | 0.9019 | 4.82955 | 0.9023 | 20.1525 | 0.0278 |
| Labor force | 124.491 | 0.0000 | 60.5693 | 0.0000 | 76.7767 | 0.0000 |
| FDI | 91.9613 | 0.0000 | 76.0335 | 0.0000 | 115.188 | 0.0000 |

Panel C. Im, Pesaran, and Shin Test: Null: Unit root (assumes individual unit root process)

| | **With Intercept** | | **With Intercept and Trend** | | **No Intercept and Trend** | |
|---|---|---|---|---|---|---|
| Variable Name | Statistic | Prob | Statistic | Prob | Statistic | Prob |
| Economic growth | −1.97739 | 0.0240 | −4.13443 | 0.0000 | | |
| Economic Freedom | −1.76214 | 0.0390 | 0.05410 | 0.5216 | | |
| Fixed Capital | −4.19965 | 0.0000 | −3.11733 | 0.0009 | | |
| Population | 0.83346 | 0.7977 | −2.16182 | 0.0927 | | |
| Labor force | −8.46023 | 0.0000 | −6.72767 | 0.0000 | | |
| FDI | −3.93529 | 0.0000 | −2.50187 | 0.0062 | | |

### 4.1.2. Results and Discussion

As indicated earlier, we proposed to undertake a numerical and graphical comparative analysis of our estimated results from OLS pooled panel kink regression and Bayesian pooled panel kink regression in the first part of our study. Scatterplots of real growth rate and each of the index regressors showed nonlinear relationships, which are a graphical indication of the presence of kinks. The plots are omitted due to limited space. To proceed, we first present in Figures 1 and 2, respectively, the trace plots of our MCMC draws for property rights and government integrity to ascertain the convergence of the Markov chains to their stationary distributions. Figure 3 displays their respective distribution plots. We observed that trace plots for all coefficients showed good mixing, stable behavior, and no presence of any unique trends. All chains indicated convergence to their stationary distribution. The plots for the other variables are omitted due to lack of space.

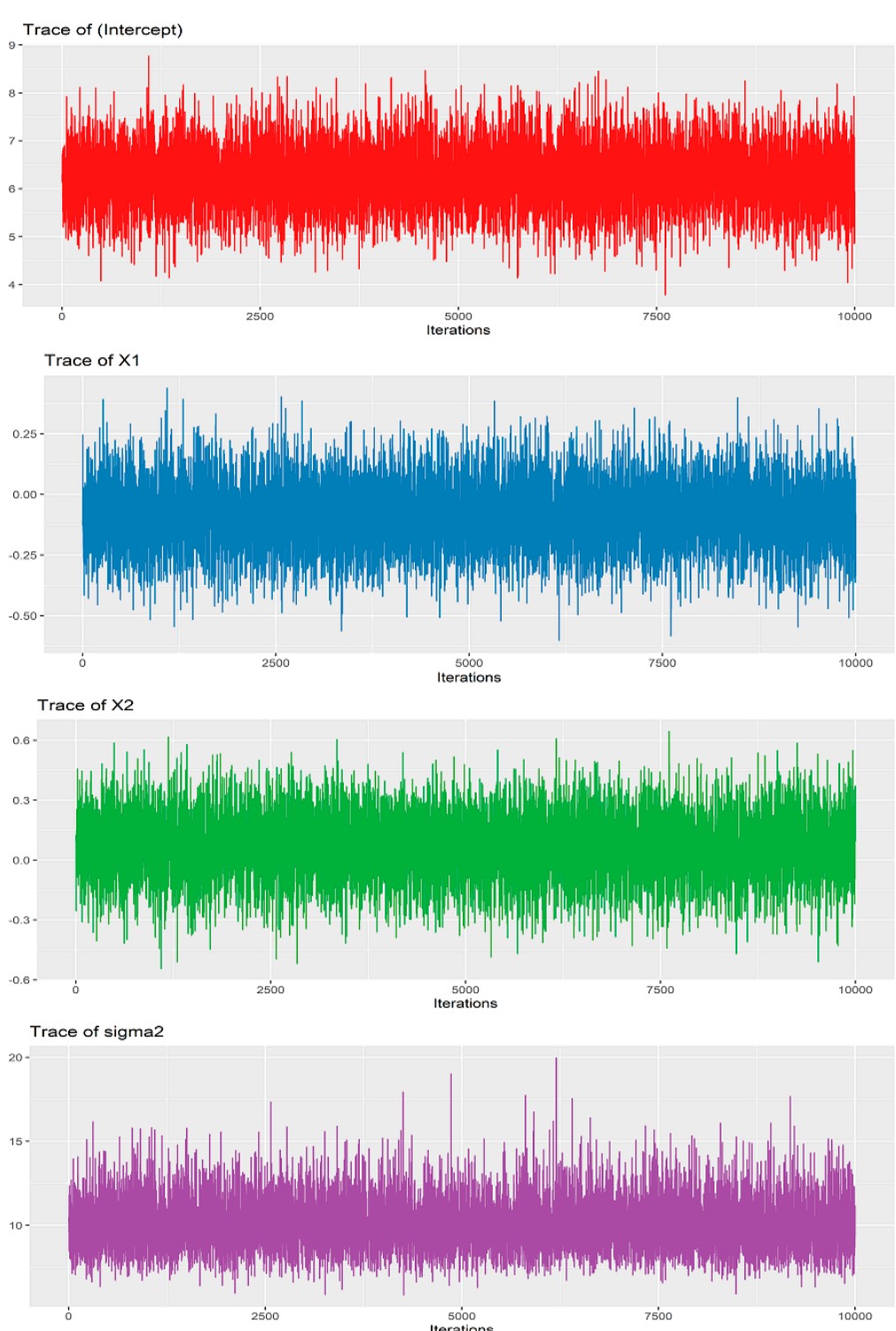

**Figure 1.** Trace plots of Property rights.

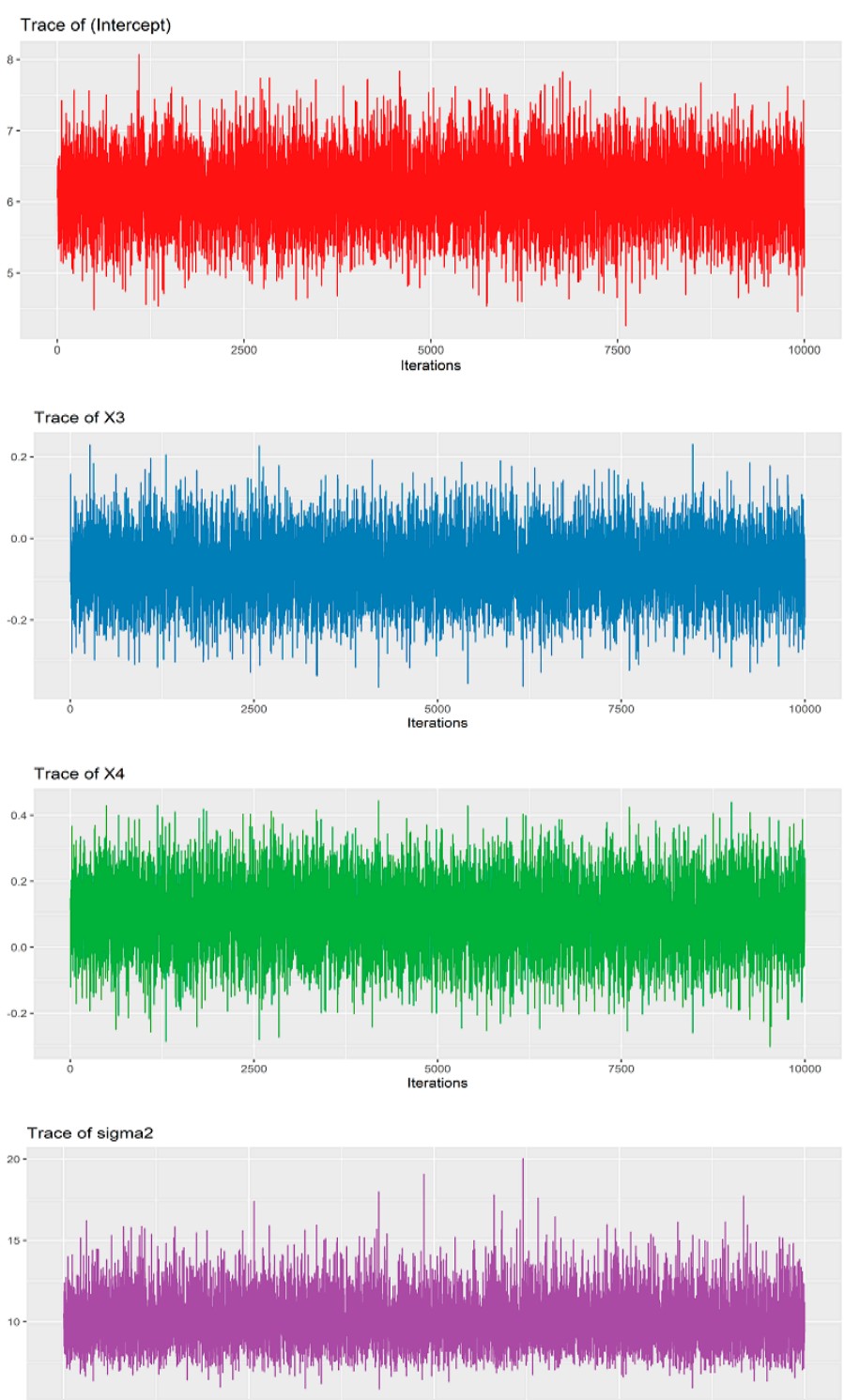

**Figure 2.** Trace plots of Government integrity.

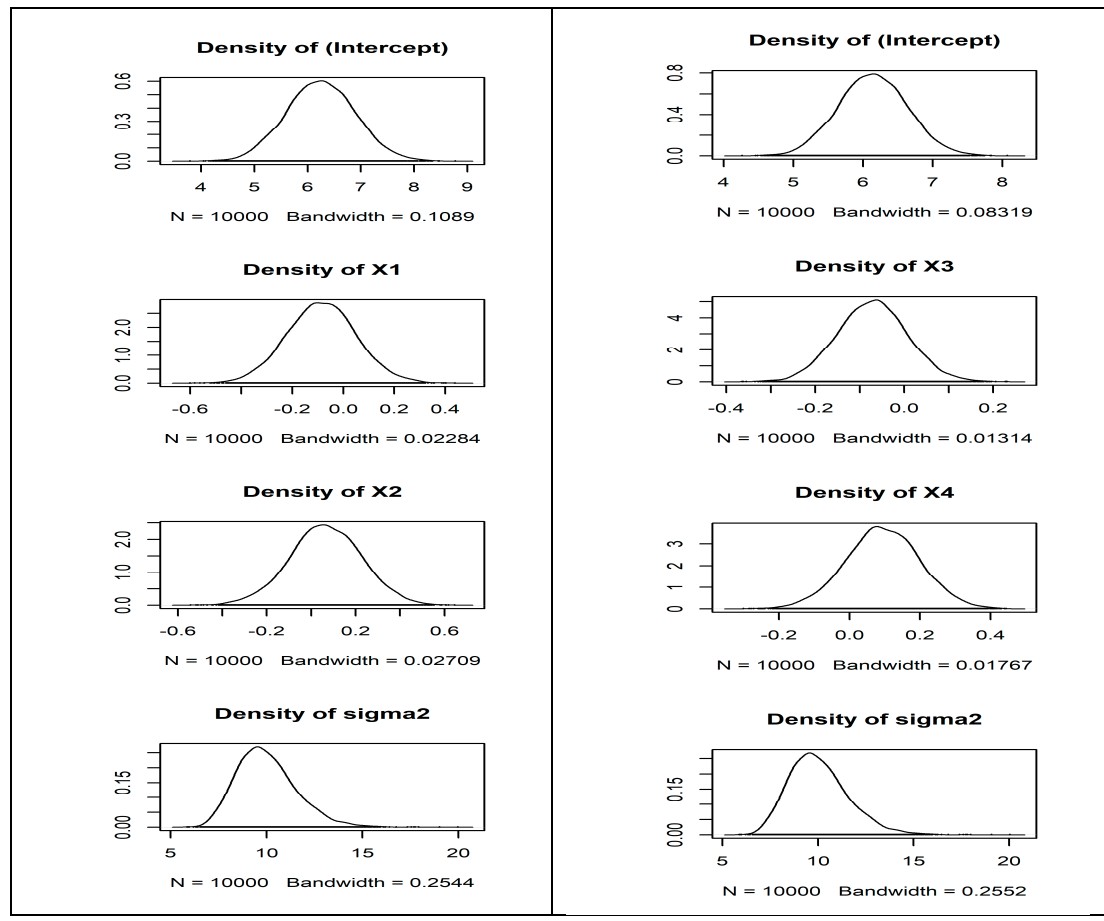

**Figure 3.** The distribution plots. **Left side**: Property rights. **Right side**: Government integrity.

The estimation results from the OLS and the Bayesian pooled panel kink regressions are presented in Tables 4 and 5, respectively. The results showed that the OLS mean estimates and the Bayesian empirical mean estimates are both very similar in all cases. For the case of the OLS framework, conditioned on the empirical mean of the index regressors, the results showed that, except tax freedom, none of the regressors had a dual statistical significance at all of the conventional levels of 1%, 5%, and 10% for the lower and upper regimes, although each intercept term for all regressors was strongly significant at 1%. One revealing observation from the OLS results was that save tax freedom, none of the OLS estimates for the upper regimes of all other indexes was significant. This was true even for those regressors whose lower regimes were statistically significant.

**Table 4.** Estimates from OLS Pooled Panel Kink Regression.

| Explanatory Variables | Estimate | Std. Error | t-Value | Pr(> \|t\|) |
|---|---|---|---|---|
| Property Right | | | | |
| Intercept | 6.25306 | 0.6437 | 9.714 | $1.55 \times 10^{-15}$ *** |
| X1 | −0.08206 | 0.1351 | −0.608 | 0.545 |
| X2 | 0.06227 | 0.1606 | 0.388 | 0.699 |
| Government Integrity | | | | |
| Intercept | 6.14631 | 0.4918 | 12.498 | $<2 \times 10^{-16}$ *** |
| X3 | −0.06989 | 0.0778 | −0.898 | 0.371 |
| X4 | 0.09184 | 0.1042 | 0.881 | 0.381 |

**Table 4.** *Cont.*

| Explanatory Variables | Estimate | Std. Error | t-Value | Pr(>\|t\|) |
|---|---|---|---|---|
| Tax Burden | | | | |
| Intercept | 8.4071 0.6092 | 13.8 | $<2 \times 10^{-16}$ | *** |
| X5 | 0.3029 | 0.062 | 4.886 | $4.65 \times 10^{-6}$ *** |
| X6 | −0.75532 | 0.20771 | −3.636 | 0.000468 *** |
| Government Spending | | | | |
| Intercept | 5.7089 | 0.5687 | 10.038 | $3.38 \times 10^{-16}$ *** |
| X7 | −0.2621 | 0.0939 | −2.79 | 0.00647 ** |
| X8 | 0.3227 | 0.201 | 1.605 | 0.1121 |
| Business Freedom | | | | |
| Intercept | 7.3474 | 0.6117 | 12.012 | $<2 \times 10^{-16}$ *** |
| X9 | 0.2672 | 0.0992 | 2.693 | 0.0085 ** |
| X10 | −0.2564 | 0.1533 | −1.673 | 0.0979 |
| Labor Freedom | | | | |
| Intercept | 6.837 | 0.5138 | 13.306 | $<2 \times 10^{-16}$ *** |
| X11 | 0.0832 | 0.0752 | 1.106 | 0.272 |
| X12 | −0.0938 | 0.1 | −0.938 | 0.351 |
| Monetary Freedom | | | | |
| Intercept | 6.5477 | 0.5156 | 12.7 | $<2 \times 10^{-16}$ *** |
| X13 | −0.1914 | 0.0798 | −2.4 | 0.0185 * |
| X14 | −0.0293 | 0.1557 | −0.188 | 0.8511 |
| Trade Freedom | | | | |
| Intercept | 6.7721 | 0.6354 | 10.658 | $<2 \times 10^{-16}$ *** |
| X15 | 0.0725 | 0.1046 | 0.693 | 0.49 |
| X16 | −0.1072 | 0.1909 | −0.562 | 0.576 |
| Investment Freedom | | | | |
| Intercept | 5.9471 | 0.5893 | 10.092 | $2.62 \times 10^{-16}$ *** |
| X17 | −0.0828 | 0.0396 | −2.092 | 0.0394 * |
| X18 | 0.077 | 0.0733 | 1.051 | 0.2961 |
| Financial Freedom | | | | |
| Intercept | 6.4209 | 0.4938 | 13.003 | $<2 \times 10^{-16}$ *** |
| X19 | −0.1251 | 0.036 | −3.47 | 0.0008 *** |
| X20 | 0.0086 | 0.0751 | 0.114 | 0.9094 |

Signif. codes: * *p* < 0.05, ** *p* < 0.01, *** *p* < 0.001.

However, the Bayesian framework offered a more suitable alternative by allowing for the use of posterior distributions. Let us take property rights for example. The median of the intercept was 6.24532. This indicated that, on average, we expect the response variable to have a value of approximately 6.24532 when all predictor variables are zero. The 95% credible interval for the intercept is approximately [5.008, 7.5275]. For X1 and X2, the median of X1 is −0.08471, and the median of X2 is 0.06519. These values indicated the central tendencies of the estimated coefficients for X1 and X2. The 95% credible intervals for X1 and X2 were approximately [−0.352, 0.1860] and [−0.260, 0.3839], respectively. The median of the variance, sigma2, was 9.84391, representing the central tendency of the estimated variance. The 95% credible interval for sigma2 was approximately [7.420, 13.5062]. This interval provided a range of plausible values for the variance term, capturing the uncertainty associated with the estimation. These intervals represented the plausible range of values for the coefficients. The results for the other variables can be similarly interpreted. This framework allowed for a probabilistic understanding of the relationships between the index regressors and economic growth, considering the uncertainty captured by the posterior distribution. For all practical intents and purposes, the Bayesian approach provided a better characterization of our data by allowing us to capture randomness in the estimates.

We presented a discussion on the causal relationship between each regressor and economic growth using graphs later.

**Table 5.** Estimates from Bayesian Pooled Panel Kink Regression.

| Variable | 2.50% | 25% | 50% | 75% | 97.50% |
|---|---|---|---|---|---|
| *Property Right* | | | | | |
| Intercept | 5.008 | 5.80939 | 6.24532 | 6.681726 | 7.5275 |
| X1 | −0.352 | −0.17665 | −0.08471 | 0.005515 | 0.186 |
| X2 | −0.26 | −0.04176 | 0.06519 | 0.174291 | 0.3839 |
| sigma2 | 7.42 | 8.89258 | 9.84391 | 10.922 | 13.5062 |
| *Government Integrity* | | | | | |
| Intercept | 5.1948 | 5.80734 | 6.1404 | 6.47383 | 7.1201 |
| X3 | −0.2267 | −0.1244 | −0.07146 | −0.01959 | 0.0853 |
| X4 | −0.1171 | 0.02436 | 0.09344 | 0.16533 | 0.3002 |
| sigma2 | 7.4441 | 8.92085 | 9.8752 | 10.95673 | 13.5492 |
| *Tax Burden* | | | | | |
| Intercept | 7.2284 | 7.9872 | 8.3998 | 8.8128 | 9.6133 |
| X5 | 0.1782 | 0.2592 | 0.3015 | 0.3428 | 0.426 |
| X6 | −1.1767 | −0.8903 | −0.7526 | −0.6104 | −0.3433 |
| sigma2 | 5.8972 | 7.0672 | 7.8232 | 8.68 | 10.7338 |
| *Government Spending* | | | | | |
| Intercept | 4.6085 | 5.3169 | 5.7021 | 6.0876 | 6.835 |
| X7 | −0.4514 | −0.3281 | −0.2638 | −0.2013 | −0.074 |
| X8 | −0.0823 | 0.1913 | 0.3259 | 0.4635 | 0.7228 |
| sigma2 | 6.8003 | 8.1494 | 9.0212 | 10.0092 | 12.3775 |
| *Business Freedom* | | | | | |
| Intercept | 6.16387 | 6.9258 | 7.34 | 7.7547 | 8.55848 |
| X9 | 0.06838 | 0.198 | 0.2652 | 0.3315 | 0.46633 |
| X10 | −0.56471 | −0.3551 | −0.2547 | −0.1487 | 0.04956 |
| sigma2 | 6.62624 | 7.9408 | 8.7903 | 9.753 | 12.06064 |
| *Labor Freedom* | | | | | |
| Intercept | 5.8429 | 6.4829 | 6.8308 | 7.1792 | 7.8543 |
| X11 | −0.0681 | 0.0304 | 0.0817 | 0.1323 | 0.2346 |
| X12 | −0.2949 | −0.1582 | −0.0927 | −0.0237 | 0.1069 |
| sigma2 | 7.4076 | 8.8772 | 9.827 | 10.9031 | 13.4829 |
| *Monetary Freedom* | | | | | |
| Intercept | 5.5501 | 6.1923 | 6.5415 | 6.891 | 7.5685 |
| X13 | −0.3521 | −0.2473 | −0.193 | −0.1398 | −0.0318 |
| X14 | −0.3427 | −0.1309 | −0.0266 | 0.0795 | 0.2804 |
| sigma2 | 6.0516 | 7.2522 | 8.0281 | 8.9073 | 11.0148 |
| *Trade Freedom* | | | | | |
| Intercept | 5.5428 | 6.3342 | 6.7645 | 7.1953 | 8.0301 |
| X15 | −0.1373 | −0.0005 | 0.0705 | 0.1407 | 0.2831 |
| X16 | −0.4931 | −0.2312 | −0.1046 | 0.0267 | 0.274 |
| sigma2 | 7.4731 | 8.9556 | 9.9137 | 10.9994 | 13.602 |
| *Investment Freedom* | | | | | |
| Intercept | 4.807 | 5.541 | 5.94 | 6.3395 | 7.1139 |
| X17 | −0.1623 | −0.1105 | −0.0837 | −0.0571 | −0.0033 |
| X18 | −0.0713 | 0.0295 | 0.0783 | 0.1281 | 0.2233 |
| sigma2 | 6.9808 | 8.3657 | 9.2607 | 10.2745 | 12.7061 |
| *Financial Freedom* | | | | | |
| Intercept | 5.4655 | 6.0806 | 6.415 | 6.7498 | 7.3987 |
| X19 | −0.1978 | −0.1504 | −0.1258 | −0.1018 | −0.0533 |
| X20 | −0.143 | −0.0405 | 0.0102 | 0.0608 | 0.1577 |
| sigma2 | 5.4066 | 6.4792 | 7.1724 | 7.9579 | 9.8408 |

We now turn to the graphs of our model fits. Figure 4 (panels (a) to (f)) and Figure 5 (panels (a) to (d)) present a pair of graphical representations of the model fits from OLS and Bayesian estimations. The colors red, blue, and green represent, respectively, the mean, lower, and upper credible bounds. The plots indicated that, although both frameworks provided fairly similar plots for the estimated means, the Bayesian credible intervals provided by the posterior distribution of all regressors offered a more appropriate capturing of the data than in the case of the OLS plots. This lent a graphical support to our earlier observations based on the numerical estimates. We now present a brief discussion of the causal relationships.

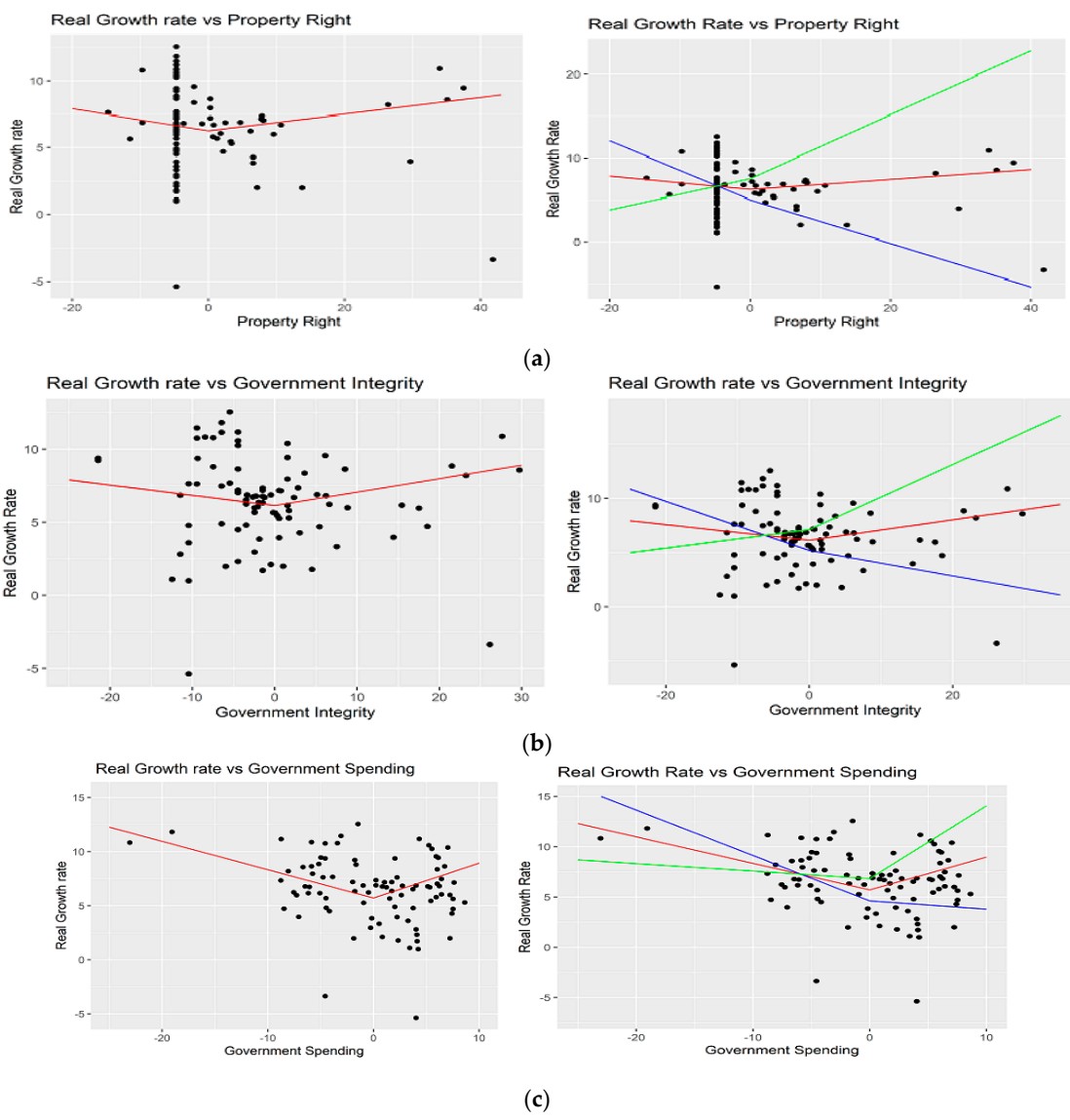

**Figure 4.** *Cont.*

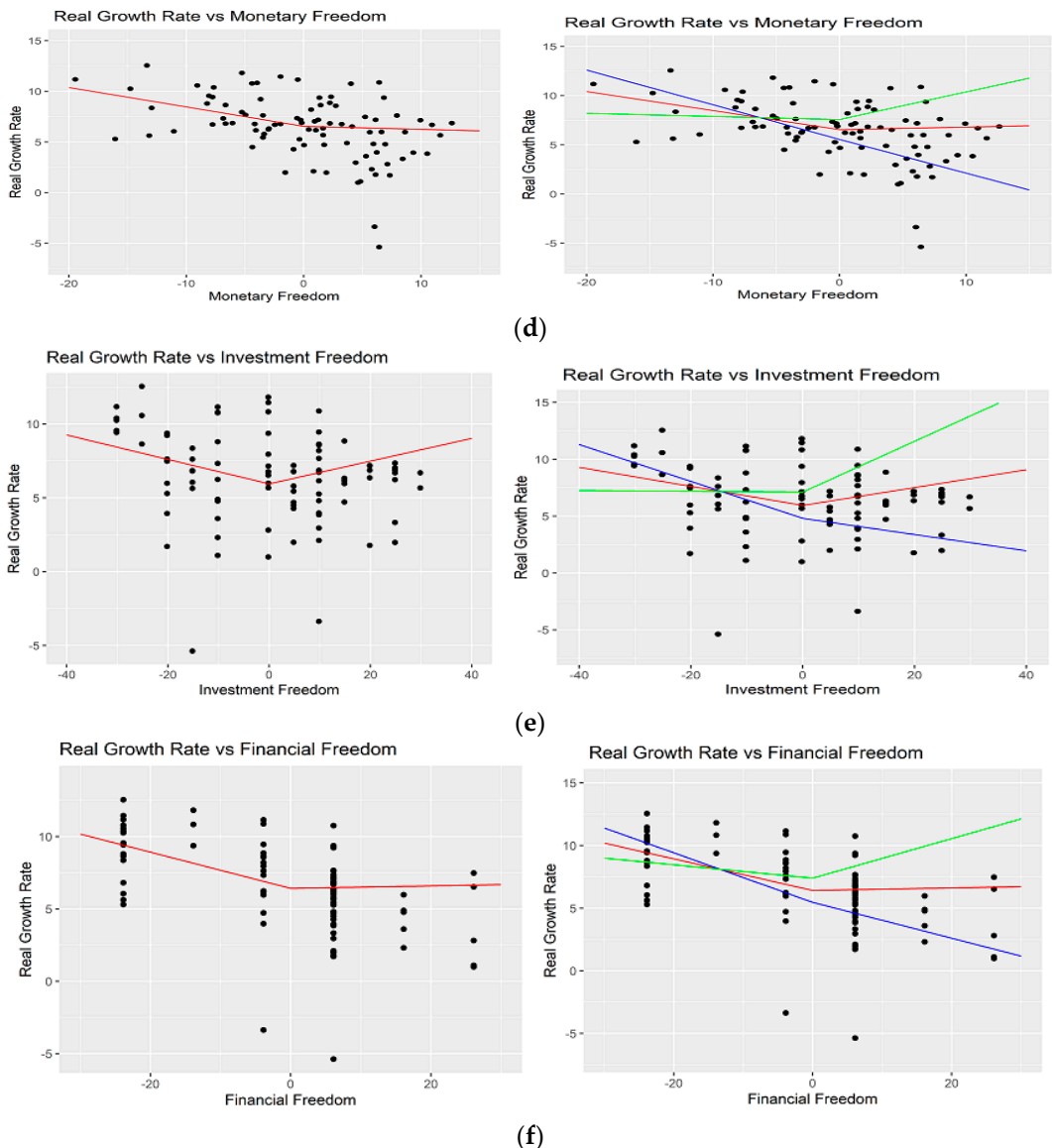

**Figure 4.** Bivariate Scatterplots with overlaid OLS and Bayesian Model Fits Plots Note: 1. Left-side figures: Classical (OLS). Right-side figures: Bayesian 2. The colors red, blue, and green represent, respectively, the mean, lower, and upper credible bounds.

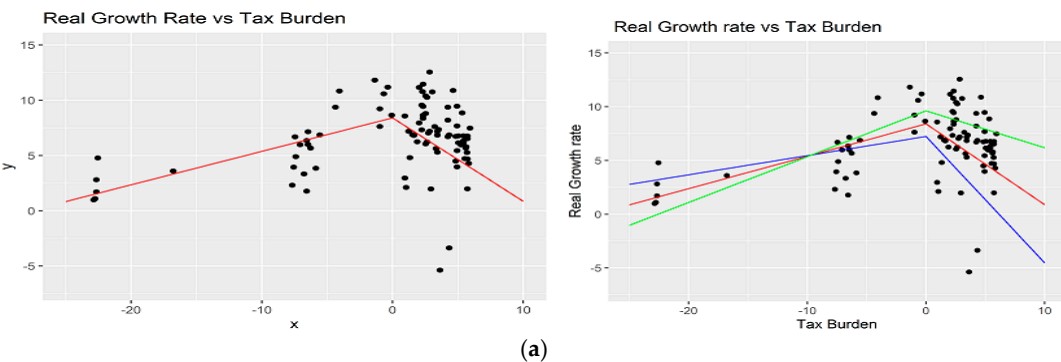

**Figure 5.** *Cont.*

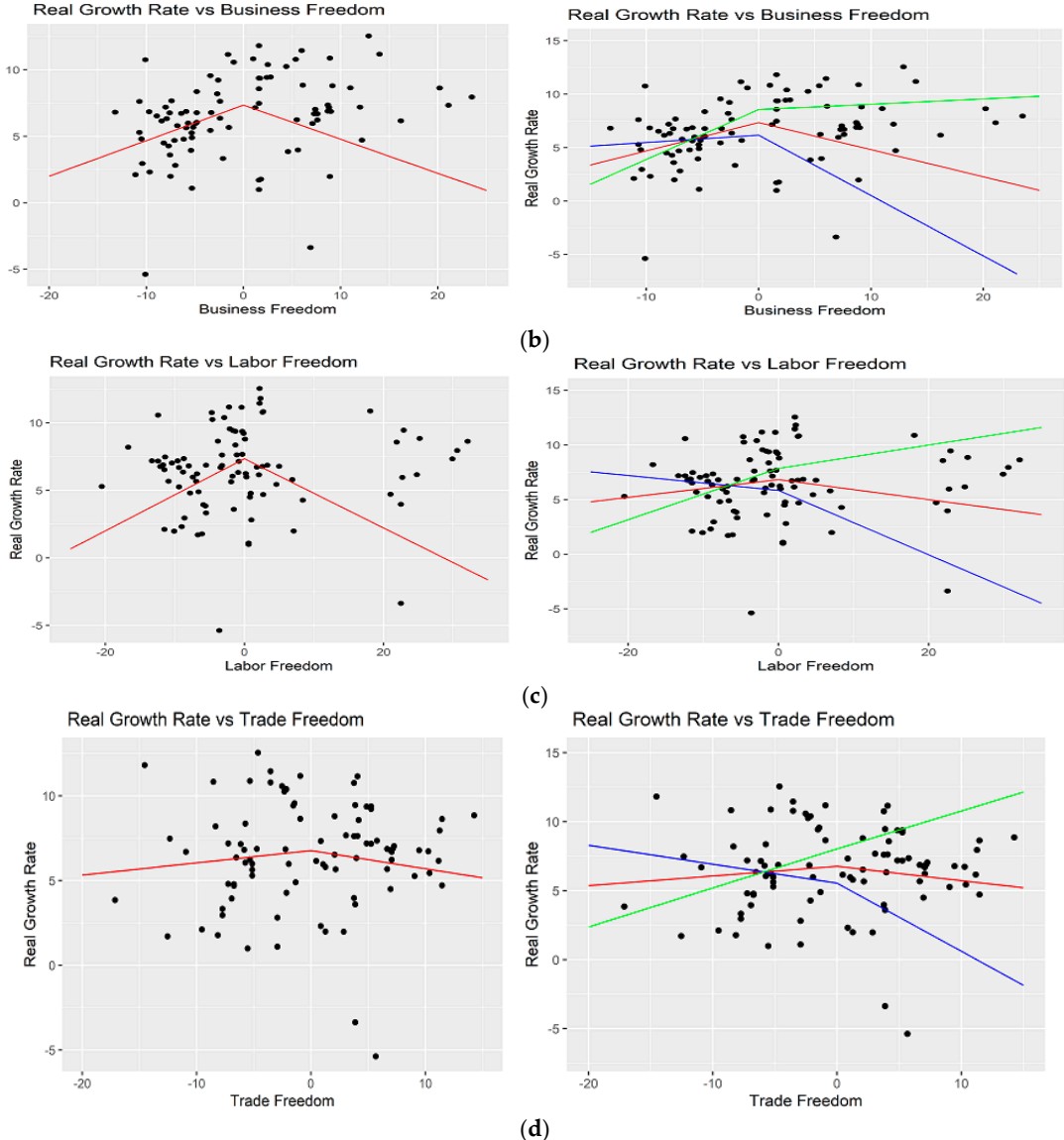

**Figure 5.** Bivariate Scatterplots with overlaid OLS and Bayesian Model Fits Plots. Note: 1. Left-side figures: Classical (OLS). Right-side figures: Bayesian. 2. The colors red, blue, and green represent, respectively, the mean, lower, and upper credible bounds.

The graphical results showed that property rights, government integrity, government spending, monetary freedom, investment freedom, and financial freedom indexes have a bivariate *convex-like* shape with economic growth. This shows that, below the threshold of their respective empirical means, each of these variables had a negative relationship with economic growth and a positive relationship beyond that point. This implies that for these indexes to positively impact economic growth, they are required to go beyond the kink points. For concreteness, a mix of policy initiatives and implantations that define each of these indexes are required to be enhanced to promote economic growth.

The above findings offer some wealth of policy considerations in the fight against geo-economic fragmentation of the Sub-Saharan African economies, which is due to the lack of strong intra-sub-regional cooperation. Worse-performing economies in the sub-region can emulate the selected economies in our study by adopting and adapting the cluster of policies that enhanced their performances on the aforementioned indexes and their rankings. In terms of the property rights index, policies that can explicitly and implicitly reduce and eliminate the risk of undue government expropriation, promote legally guaranteed

intellectual property rights, and ensure the quality of contract enforcement would be required to be rolled out. In the case of the government integrity index, key policies that deal explicitly with corruption perceptions, bribery risks, and state capture by elite groups must be implemented as a matter of policy.

As part of the policy options to score highly on the government spending index scale, streamlining all government expenditures to strictly exclude all forms of profligate expenditures, including transfer payments of all kinds that fail the litmus test of cost–benefit considerations, would be required. One index that requires immediate policy attention due to the current inflationary pressures in the sub-region is the monetary index. Government interventions and involvements that grossly distort price stability in factor and commodity markets require policy realignments that promote macro-stability along with zero, or extremely limited, government microeconomic policy interventions. The same goes for both investment and financial indexes. Policy initiatives would be required to ease unwarranted investment restrictions on both local and foreign capital inflows. Cumbersome and inefficient bureaucracies that characterize investment procedures in most economies within the sub-region would require urgent policies to remove them. Government policy initiatives to ensure sound macro-prudential environments must be rigidly guided and guarded to avoid overly restrictive financial sector regulations so as not to stifle financial sector development, with a possible adverse impact on economic growth.

The graphical model fits for tax burden, business freedom, labor freedom, and trade freedom indexes show *concave-like* results, indicating that lower regimes of these indicators promote economic growth and that they reduce it in the upper regime (see Figure 5 (panels (a) to (d))). This finding is diametrically opposite to the earlier set of indexes discussed above. This latter finding indicates that these freedom indicators should not be allowed to go beyond certain thresholds.

Our study indicated that policy choices that push the above indexes above their empirical mean levels should be discouraged. For instance, in the case of the tax burden index, a mix of policies that keep the score from going beyond the mean should be rigorously pursued. Summarily, the total tax burden as a proportion of GDP should not be allowed to drop via the easing of individual and corporate income tax rates. This is an indication that the current tax regimes in the selected economies are appropriate for economic growth. In the case of labor freedom, unbounded labor freedom has some consequences for productivity and participation rates. Given this, labor-related policies must be directed at achieving outcomes that ensure productivity and labor rewards. These can be achieved through the implementation of realistic minimum wage rates, flexible self-development policies, and conductive non-monetary benefits. In the case of trade freedom, policy choices would be required to be designed in a fashion not to overly regulate and unguardedly liberalize trade. Some minimal amounts of quantitative, regulatory, and customs restrictions would be required to sanitize the trade environment, as overly liberalized trade could have negative repercussions for economic growth, just as excessive trade controls deny economies the opportunity of reaping trade benefits, noting that single economy can claim complete autarky.

### 4.2. Results and Discussions of Threshold and Bayesian Models

The estimation results from the pooled panel threshold and the Bai–Perron multiple threshold test are presented in Tables 6 and 7, respectively. First of all, it is important to highlight that both frameworks yielded completely different parameter estimates despite the use of the same numerical threshold value, unlike in the case of part one of our analysis. The results of the Bai–Perron multiple threshold test in Table 7 showed that economic freedom has only a single threshold value, which occurred at 56.70, and it is strongly significant at the 0.05 level. In low regimes, economic freedom has a negative impact on growth. Specifically, a 1 percentage point increase in economic freedom score reduces growth by approximately 0.97 percentage points with a significance at the 0.01 level, with all other things being equal. The parameter estimate for the high regime yields a

counterintuitive sign; however, it is not statistically different from zero. In the exception of fixed capital, which has the a priori sign and is significant at the 0.01 and 0.05 levels for the low and high regimes, respectively, all other factors are not statistically different from zero in both regimes, although they all have their expected signs. In the lower regime, a unit percentage point rise in the rate of fixed capital accumulation as a percentage of GDP led to approximately a 0.0867 percentage point increase in the growth rate, whereas it was 0.0863 in the upper regime. Population, labor force, and FDI had coefficients of, respectively, 1.9092, 0.0786, and 0.2138 in the lower regime. In the upper regime, respectively, they had coefficients of −1.43888, 0.0005, and 0.3013.

**Table 6.** Estimates: Threshold Regression.

| Variable Name | Coefficient | Std. Error | t-Statistic | Prob. |
|---|---|---|---|---|
| Economic freedom < 56.699999 | | | | |
| C | 46.5474 | 13.09041 | 3.555839 | 0.0006 |
| Economic Freedom | −0.97099 | 0.208018 | −4.667811 | 0.0000 |
| Fixed Capital | 0.086684 | 0.027064 | 3.20296 | 0.002 |
| Population | 1.909193 | 1.509761 | 1.264566 | 0.2098 |
| Labor force | 0.078563 | 0.061823 | 1.27077 | 0.2076 |
| FDI | 0.213812 | 0.312258 | 0.684729 | 0.4955 |
| 56.699999 <= Economic freedom | | | | |
| C | 12.22298 | 11.15441 | 1.095798 | 0.2765 |
| Economic Freedom | −0.060652 | 0.13295 | −0.456202 | 0.6495 |
| Fixed Capital | 0.086271 | 0.036853 | 2.340942 | 0.0218 |
| Population | −1.438804 | 1.770718 | −0.812554 | 0.4189 |
| Labor force | 0.000521 | 0.053889 | 0.009667 | 0.9923 |
| FDI | 0.301326 | 0.356649 | 0.844882 | 0.4008 |
| R-squared | 0.449264 | Mean dependent var | | 6.498889 |
| Adjusted R-squared | 0.371596 | S.D. dependent var | | 3.175414 |
| S.E. of regression | 2.517212 | Akaike info criterion | | 4.807747 |
| Sum squared resid | 494.2357 | Schwarz criterion | | 5.141055 |
| Log likelihood | −204.3486 | Hannan−Quinn criter. | | 4.942156 |
| F-statistic | 5.784429 | Durbin−Watson stat | | 1.865064 |
| Prob(F-statistic) | 0.000001 | | | |

**Table 7.** Multiple Threshold Tests (Bai–Perron tests of L + 1 vs. L Sequentially Determined Thresholds).

| Threshold Test | F-Statistic | Scaled F-Statistic | Critical Value ** |
|---|---|---|---|
| 0 vs. 1 * | 5.520436 | 33.12261 | 20.08 |
| 1 vs. 2 | 2.281156 | 13.68694 | 22.11 |

* $p < 0.05$, ** $p < 0.01$.

The estimation results from the Bayesian pooled panel kink regression are represented in Tables 8 and 9 below. With a total of 10,000 iterations and 10% burn-in, the trace and density plots of all variables show good mixing, no unique trends, and convergence to their stationary distribution. The plots are omitted due to lack of enough space. Our results showed that the sign of the impact of economic freedom on growth in the low regime is the same as in the case of the threshold estimates, albeit with a different marginal impact. Specifically, the results indicated that in the low regime, a 1 percentage point rise in economic freedom will induce about a 0.24 percentage point decline in economic growth, ceteris paribus, with a 95% probability. This finding agrees with the propositions of Zahonogo (2017), Bergh and Nilsson (2010), Carter (2007), and Kneller et al. (1999). It also lent support to the earlier findings by Santiago et al. (2020), Justesen (2008), and Carlsson and Lundström (2002).

**Table 8.** Bayesian Estimates.

| Variable Name | Mean | SD | Naive SE Time | Time Series SE |
|---|---|---|---|---|
| Intercept | 16.655386 | 9.61783 | 0.0961783 | 0.0961783 |
| Econ freedom (low regime) | −0.244083 | 0.13796 | 0.0013796 | 0.0013796 |
| Econ freedom (high regime) | 0.02007 | 0.02004 | 0.0002004 | 0.0002004 |
| Fixed Capital | 0.078552 | 0.02511 | 0.0002511 | 0.0002511 |
| Population | 0.42296 | 1.324 | 0.01324 | 0.01324 |
| Labor force | 0.002014 | 0.04663 | 0.0004663 | 0.0004663 |
| FDI | 0.509866 | 0.26687 | 0.0026687 | 0.0026687 |
| sigma2 | 8.593133 | 1.35955 | 0.0135955 | 0.0148669 |

**Table 9.** Bayesian Estimates: Quantiles for each Variable.

| Variable Name | 2.50% | 25% | 50% | 75% | 97.50% |
|---|---|---|---|---|---|
| Intercept | −2.21649 | 10.277606 | 16.62382 | 23.0581 | 35.75165 |
| Econ freedom (low regime) | −0.51599 | −0.337402 | −0.24319 | −0.14979 | 0.02395 |
| Econ freedom (high regime) | −0.0185 | 0.006736 | 0.019719 | 0.03367 | 0.05942 |
| Fixed Capital | 0.02943 | 0.061624 | 0.07867 | 0.0953 | 0.12833 |
| Population | −2.20742 | −0.447318 | 0.424131 | 1.30976 | 3.02618 |
| Labor force | −0.08898 | −0.029399 | 0.002294 | 0.03347 | 0.09461 |
| FDI | −0.02091 | 0.328214 | 0.511734 | 0.68866 | 1.03107 |
| sigma2 | 6.32842 | 7.635171 | 8.453642 | 9.38782 | 11.6283 |

In contrast to the threshold regression estimates, the Bayesian estimates showed that in the high regime, economic freedom has a positive impact on growth. Specifically, with a 95% probability, a 1 percentage point rise in economic freedom would cause growth to rise by approximately 0.02 percentage points. This finding seemed to lend some support to the propositions of Gertz and Evers (2020), Baharumshah et al. (2016), Feenstra (2015), and Mitra et al. (2014). This also confirmed the surveys of Hall and Lawson (2014) and Doucouliagos and Ulubasoglu (2006) and the findings of Krueger (1998) and Kacprzyk (2016). By establishing a nonlinear impact of economic freedom on growth, the above findings from our study are novel in the economic growth–economic freedom literature.

## 5. Conclusions

Employing our pooled panel kink modeling technique under the condition of a known policy index choice of economic freedom, we found that the Bayesian framework is more appropriate than the OLS approach. The former provides the possibility of accounting for randomness in the parameter estimates by using probabilistic posterior distributions. It is important to highlight that this has two main contributions to the existing literature. As evident in our literature review, there are numerous studies on kink regression with all its variants employing unbounded, compact, and continuous explanatory variables with their fine statistical properties. Our study is the first to employ bounded index regressors with kink regression, and this is novel.

Our findings indicated that the impact of economic freedom on economic growth is contingent on the policy choice regimes, and, therefore, economic freedom has a nonlinear impact on economic growth. This finding shows that wholesale adoption and implementation of policies and measures that seek to promote economic freedom should not be carried out, as there is evidence that beyond some levels, excessive economic freedom could harm growth. This finding is significantly different from earlier studies that find either a negative or a positive impact of economic freedom on economic growth without analyzing the possibilities of kink effects. Our further application of both threshold and Bayesian pooled panel regressions has lent added support to our earlier findings. Although the signs and impacts for the high regime differ under both frameworks, our results confirm a negative impact of economic freedom on growth in low regimes. This prompts several policy directions for Sub-Saharan countries to chart.

For future research, we suggest a fixed-effects Bayesian kink regression technique that will apply panel kink regression using panel data on economic freedom indicators for a mix of low- and high-performing economies within the sub-region to account for country-specific heterogeneities, as our present study focused on pooled panel analysis.

**Author Contributions:** Conceptualization, E.M. and C.C.; methodology, E.M. and C.C.; software, E.M. and C.C.; validation, E.M., C.C., K.C. and S.S.; formal analysis, E.M.; investigation, E.M.; resources, E.M. and C.C.; data curation, E.M.; writing—original draft preparation, E.M.; writing—review and editing, E.M. and C.C.; visualization, E.M.; supervision, C.C., K.C. and S.S.; project administration, E.M.; funding acquisition, E.M., C.C., K.C. and S.S. All authors have read and agreed to the published version of the manuscript.

**Funding:** This research received no external funding.

**Institutional Review Board Statement:** Not applicable.

**Informed Consent Statement:** Not applicable.

**Data Availability Statement:** Not applicable.

**Acknowledgments:** This research work was partially supported by Chiang Mai University.

**Conflicts of Interest:** The authors declare no conflict of interest.

**Appendix A**

Define the true values. Assume the true parameter values for $(\beta, \delta)$ are given by

$$\left(\beta^{0\prime}, \delta^0\right)' = \underset{\beta \in \mathcal{B}, \delta \in \Delta}{\mathrm{argmin}} L(\beta, \delta) \tag{A1}$$

where $L(\beta, \delta)$, the demeaned loss function, is defined as

$$L(\beta, \delta) = \mathrm{E}\|Y_i - X_i(\delta)\beta\|^2_{M_{t_T}} = \sum_{t=1}^{T} \mathrm{E}\left[\widetilde{Y}_{it} - \widetilde{X}'_{it}(\delta)\beta\right]^2 \tag{A2}$$

where $\widetilde{Y}_{it} = Y_{it} - \overline{Y}_i$, $\overline{Y}_i = T^{-1}\sum_{t=1}^{NT} Y_{it}$, $\widetilde{X}(\delta) = X_{it}(\delta) - \overline{X}_i(\delta)$, and $\overline{X}_i(\delta) = T^{-1}\sum_{t=1}^{T} X_{it}(\delta)$. For any given $\delta \in \Delta$, objective function then becomes

$$L^c(\delta) = L(\beta(\delta), \delta) = \sum_{t=1}^{T} \mathrm{E}\left[\widetilde{Y}_{it} - \widetilde{X}'_{it}(\delta)\beta(\delta)\right]^2 \text{ with}$$

$$\beta(\delta) = \left(\sum_{t=1}^{T} \mathrm{E}[\widetilde{X}(\delta)\widetilde{X}'_{it}(\delta)]\right)^{-1} \sum_{t=1}^{T} \mathrm{E}\left[\widetilde{X}'_{it}(\delta)Y_{it}\right]$$

$$\left.\begin{array}{l} L^c(\delta)L(\beta(\delta), \delta) = \sum\limits_{t=1}^{T} \mathrm{E}\left[\widetilde{Y}_{it} - \widetilde{X}'_{it}(\delta)\beta(\delta)\right]^2 \\[2mm] \beta(\delta) = \left(\sum\limits_{t=1}^{T} \mathrm{E}[\widetilde{X}(\delta)\widetilde{X}'_{it}(\delta)]\right)^{-1} \sum\limits_{t=1}^{T} \mathrm{E}\left[\widetilde{X}'_{it}(\delta)Y_{it}\right] \end{array}\right\} \tag{A3}$$

Under the combined assumption of the existence of a global minimum and concentration property (see Zhang et al. (2017) for a detailed exposition on the key assumptions), the true value of $\delta$ is given by

$$\delta^0 = \mathrm{argmin}_{\delta \in \Gamma} L^c(\delta). \tag{A4}$$

Denote $1_{it}^+(\delta) = 1(X_{1,it} > \delta)$ and $i_{it}^+(\delta) = 1_{it}^+(\delta) - \frac{1}{T}\sum_{t=1}^T 1_{it}^+(\delta)$, and $1_{it}^-(\delta) = 1(X_{1,it} < \delta)$ and $i_{it}^-(\delta) = 1_{it}^-(\delta) - \frac{1}{T}\sum_{t=1}^T 1_{it}^-(\delta)$, and let $e_{it}(\theta) = \widetilde{Y}_{it} - \widetilde{X}_{it}'(\delta)\beta$,

$$e_i(\theta) = (e_{i1}(\theta), \ldots, e_{iT}(\theta))' \; f_{it}(\theta) = -\frac{\partial e_{it}(\theta)}{\partial\theta} = \begin{pmatrix} \widetilde{X}_{it}'(\delta)(\delta) \\ -\beta_1^- i_{it}^-(\delta) - \beta_1^+ 1_{it}^+(\delta) \end{pmatrix},$$

$$f_i(\theta) = (f_{i1}'(\theta), \ldots, f_{iT}'(\theta))', \text{ and}$$

$$D_{it}(\delta) = -\frac{\partial}{\partial\theta'}f_{it}(\theta) = \begin{pmatrix} 0 & 0 & 0_{1\times d} & \dot{1}_{it}^-(\delta) \\ 0 & 0 & 0_{1\times d} & i_{it}^+(\delta) \\ 0_{d\times 1} & 0_{d\times 1} & 0_{d\times d} & 0 \\ i_{it}^-(\delta) & i_{it}^+(\delta) & 0_{1\times d} & 0 \end{pmatrix}.$$

From Equation (5), it can be shown that the first derivatives of $Q_{NT}(\beta, \delta)$ is $S_N(\theta) = \frac{-1}{N}\sum_{i=1}^N f_i'(\theta)e_i(\theta)$, where the second is $F_N(\theta) = \frac{1}{N}\sum_{i=1}^N \left[f_i'(\theta)f_i(\theta) + \sum_{t=1}^T D_{it}(\delta)e_{it}(\theta)\right]$. Denote $f_{it} = f_{it}(\theta^0)$, $f_i = f_i(\theta^0)$, $e_{it} = e_{it}(\theta^0)$, $e_i = e_i(\theta^0)$, and $D_{it} = D_{it}(\delta^0)$. To establish the asymptotic distribution of $\hat{\theta}$, Zhang et al. (2017) posit five key assumptions, which include following:

**Assumption A1.** *i.* $(Y_i, X_i)$ *are independently identically distributed (i.i.d.) across i; (ii)* $E|Y_{it}|^{4+\delta} < \infty$ *and* $E\|X_{it}\|^{4+\delta} < \infty$ *for some* $\delta > 0$*; and (iii)* $E(u_{it}) = 0$ *and* $E|u_{it}|^{4+\delta} < \infty$.

**Assumption A2.** *(i)* $Z_{it}$ *has a probability density function (pdf) given by* $f_{z,t}(z)$*, where* $Z_{it} \equiv (X_{it}, Y_{it})$*; (ii)* $(Z_{it}, Z_{is})$ *has a joint PDF* $f_{z,ts}$*; and (iii)* $f_t(x)$ *satisfies* $\max_{1\leq t\leq T} f_t(x) \leq \overline{f} < \infty$*, where* $f_t(x)$ *is the marginal pdf of* $X_{1,it}$.

**Assumption A3.** $Q_T(\delta) \equiv E\left[\widetilde{X}_i'(\delta)\widetilde{X}_i(\delta)\right]$ *is non-singular for all* $\delta \in \Gamma \equiv [\underline{\delta}, \overline{\delta}] \subset \mathbb{R}$.

**Assumption A4.** $\beta_1^+ - \beta_1^-$ *is a constant, and* $\beta \in \mathcal{B} \subset \mathbb{R}^{d+2}$*, where* $\mathcal{B}$ *is compact.*

**Assumption A5.** $\delta^0 = argmin_{\delta\in\Gamma} L^c(\delta)$ *is unique.*

Under Assumptions $1-5$, as $N \to \infty$,

$$\left.\begin{array}{r} \sqrt{N}(\hat{\theta} - \theta^0) \xrightarrow{d} N(0, V) \\ \text{where } V = G^{-1}SG^{-1} \\ S = \text{Var}(h_i'e_i) \\ G = E[H_N(\theta^0)] = E(h_i', h_i) + \sum_{t=1}^T E(D_{it}, e_{it}) \end{array}\right\} \quad (A5)$$

The asymptotic covariance matrix $V$ can be estimated using $\hat{V}_N = \hat{G}_N^{-1}\hat{S}_N\hat{G}_N^{-1}$, where

$$\hat{S}_N = \frac{1}{N}\sum_{i=1}^N h_i'(\hat{\theta})e_i(\hat{\theta})e_i'(\hat{\theta})h_i(\hat{\theta}) \text{ and } \hat{G}_N = \frac{1}{N}\sum_{i=1}^N\sum_{t=1}^T [h_{it}'(\hat{\theta})h_{it}(\hat{\theta}) + D_{it}(\hat{\delta})e_{it}(\hat{\theta})].$$

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
