# Peer review of "Comparing Classical and Bayesian Panel Kink Regression Frameworks in Estimating the Impact of Economic Freedom on Economic Growth"

_economies, doi:10.3390/economies11100253_

Round 1

Reviewer 1 Report

Here are my comments of the paper, Comparing Classical and Bayesian Panel Kink Regression Frameworks in Estimating the Impact of Economic Freedom on Economic Growth submitted to Economies.

1.  The introduction is quite plodding.  The author(s) need to separate the introduction and the review of the literature.  This will make it easier for the readers to follow.  Also at the end of the review of the literature the author(s) should provide a brief summary of the review of the literature. 

2. The literature should be written in past tense as the work has already been completed; similarly with the methods and the empirical results of this paper.

3. I think the analytical details provided on pages 6 and 7 should be relegated to the appendix of the paper. This discussion hinders the empirical development of the paper for the reader. 

4. For the posterior probability, how was this determined or estimated in this paper?  The posterior probability is the most difficult part of the Bayesian estimation. 

5. On line 397 instead of make use  should be rewritten as used 

6.  Line 408 Table 1:  Descriptive Statistics of Variable should be Table 1:  Descriptive Statistics of the Variables

7. The author(s) used the Levin, Lin and Chu (2002) test for the unit root for this panel regression.  Why was this unit root test used but not other unit root tests such as Im–Pesaran–Shin (2003), and Fisher-type (Choi 2001) tests? 

8.  Line 440 scatter plots should be scatterplots

9. Are the standard errors used in the empirical results robust? The reason for robust standard errors in panel data is because the idiosyncratic errors can have heteroskedasticity or autocorrelation, or both. That is the composite error term is u(i) + e(i,t). Now, pooled OLS leaves u(i) in the error term, which is an obvious source of autocorrelation. But e(i,t) can be autocorrelated. And both u(i) and e(i,t) can both have heteroskedasticity. Clustering would handle all of these issues.  More important, in the fi8xed effect case, u(i) is removed. But e(i,t) is still there. It is still important to want your inference robust to heteroskedasticity and autocorrelation. 

10. Line 481 the first word of the sentence needs to be capitalized. 

11. Line 486, take out the fact 

12.  Line 494 graphics should be graphs 

13. 

Extensive editing of English language required

Author Response

For Reviewer 1:
1. A summary of the reviewed literature has been provided on page 4. The introduction section is on pages 1 and 2. The literature review section is on pages 3 and 4. However, we are quite unclear with the suggestion to separate the introduction from the literature review section. Please, we welcome any further directions in this regard.
2. The suggestion had been effected. The new version is highlighted in yellow.
3. The analytical details have been moved to pages 21 to 22 under Appendix 1 [from lines 667 to 697].
4. Please, the estimation of the posterior probabilities is captured on page 7, from line 305 to 322, and page 8, from line 323 to 330. The posterior probabilities were determined using the lower and upper quantiles of 2.5% and 97.5% generated from the Bayesian estimation using MCMC simulations.
5. Correction effected on page 9, paragraph 1, line 367.
6. Correction effected on page 9, line 378.
7. Revision had been effected on page 10, Table 3.
8. We effected the correction on page 10, line 400, and also on pages 16 and 18, lines 470 and 539, respectively.
9. Thanks very much for this insight. We will use a fixed-effects modeling in a follow-up paper to address this concern, please.
10. The lowercase used was due to a wrong paragraphing. The correction effected on page 13, line 441.
11. The correction is effected on page 13, lines 444 and 445.
12. The correction is effected on page 14, line 448.

Reviewer 2 Report

This is an interesting paper from a methodological perspective and the authors seem to have undertaken the technical work competently. I do however have some concerns about the work.

Why is pooled estimation performed rather than fixed or random estimation?

It is supprising that the authors choose 5 countries which have exhibited high growth rates. This can hardly be seen as a representative sample in a growth study. Why not a much larger sample of countries? This is particularly true given (13) where as n goes to infinity the authors derive the properties of the estimator but n is only 5.

The comparison between the Bayesian and the classical approach seems odd. The authors state that ' The Bayesian approach provides a more accurate fit for our data'. I would sugest that this statement must be wrong, by definition the classical approach is minimising the squared residuals and so it must give the best fit. The Bayesian approach is mixing the best fit with the proiors and must fit worse. It may be true that the bayesian approach gives more reasonable results or that it is better in some other sense but it can not fit better.

There is also a general problem of ommited regresors. Surely growth is not only a fucntion of freedom. What about capital investment, education, labour supply, foreign direct investment etc.

Author Response

  1. For Reviewer 2:
  • We derived our motivation to use pooled estimation to account for the seemingly homogenous growth records of the selected economies during the periods that preceded the COVID-19 pandemic and their demonstrated resilience during and after the peak of the pandemic. We sought to find out if other non-performing economies within the sub-region could learn from the selected economies, as regards their macroeconomic policy choices that bother on economic growth and economic freedom.

  • Please, same as point 2 above. Regarding the asymptotic properties of the estimators, we took the view that as equation (13), which has become equation (19) under Appendix 1, fulfils the requirement for convergence in distribution to a normal distribution, which requires that . Additionally, we intend to employ fixed-effects and random effects estimations in a later study, in which case a mix of top performing and less performing economies within the sub-region will be considered so as to examine the dynamics of the within and between country heterogeneities.

  • The correction has been effected, please. The statement should have read “……a better characterization of our data by allowing us to capture randomness in the estimates”. [Please, refer to page 13, lines 443 and 444].

  • We resolved this by including other regressors in the second part of our estimation [please, refer to pages 18 to 20, Tables 5a, 6a, and 6b].

Reviewer 3 Report

This is an interesting and topical study of economic freedom and economic growth using a panel kink regression framework, finding that economic freedom has negative effects below the threshold but positive above, when using the Bayesian approach. I have the following comments:

- Some parts could do with some more explanation, such as how does the kink regression resolve the endogeneity problem ? (line 306)

- In the panel regressions, using OLS, were fixed effects included?

- In the panel unit root tests, you should also do the IPS unit root tests as well as the LLC tests.

- In the regression models, it might be worth mentioning more on the effects of the control variables.

- In the discussion of the results, there should be more comparison of the results with those from other similar studies.

Minor Point:

- The paper needs to be proof read as there are some typos, such as some citations needing the date added, for instance Zhang et al.

- The references need to be consistent, i.e. Hansen...

The English language is fine, but could do with a final proof reading.

Author Response

  1. For Reviewer 3:

  • Unfortunately, our source, Maneejuk and Yamaka (2020), could not offer any detailed accounts on this. If accepted, we may have to remove this part for lack of explanation, please.

  • In this paper, we derived our motivation to use pooled estimation to account for the seemingly homogenous growth records of the selected economies during the periods that preceded the covid-19 pandemic and their demonstrated resilience during and after the peak of the pandemic. We sought to find out if other non-performing economies within the sub-region could learn from the selected economies, as regards their macroeconomic policy choices that bother on economic growth and economic freedom. We, however, intend to employ fixed-effects and random effects estimations in a later study, in which case a mix of top performing and less performing economies within the sub-region will be considered so as to examine the dynamics of the within and between country heterogeneities.

  • Revision has been effected on page 10, Table 3, Panels A, B, and C.

  • The revised version is on page 18, lines 552 to 561.

  • Please, refer to pages 19 and 20, lines 604 to 618.

Minor Point:

  • The correction is effected on page 22, line 673.
  • The correction is effected on page 23, line 721.
  •  

Round 2

Reviewer 2 Report

none